# Prevalence of Arrhythmia in Adults after Fontan Operation

**DOI:** 10.3390/jcm11071968

**Published:** 2022-04-01

**Authors:** Magdalena Okólska, Grzegorz Karkowski, Marcin Kuniewicz, Jacek Bednarek, Jacek Pająk, Beata Róg, Jacek Łach, Jacek Legutko, Lidia Tomkiewicz-Pająk

**Affiliations:** 1Cardiological Outpatient Clinic, Department of Cardiovascular Diseases, John Paul II Hospital, 31-202 Krakow, Poland; magdaokolska@gmail.com (M.O.); beatarog@interia.pl (B.R.); 2Department of Electrocardiology, Institute of Cardiology, Faculty of Medicine, Jagiellonian University Medical College, John Paul II Hospital, 31-202 Krakow, Poland; gkarkowski@interia.pl (G.K.); kuniewiczm@gmail.com (M.K.); bednarekj1@gmail.com (J.B.); 3Department of Anatomy, Jagiellonian University Medical College, 31-008 Krakow, Poland; 4Institute of Medical Sciences, Department of Surgery, Medical College of Rzeszow University, 35-959 Rzeszow, Poland; jacekpajak@poczta.onet.pl; 5Department of Cardiac and Vascular Diseases, Institute of Cardiology, Jagiellonian University Medical College, John Paul II Hospital, 31-202 Krakow, Poland; djholter@interia.pl; 6Clinical Department of Interventional Cardiology, John Paul II Hospital, 31-202 Krakow, Poland; jacek.legutko@uj.edu.pl; 7Department of Interventional Cardiology, Faculty of Medicine, Institute of Cardiology, Jagiellonian University Medical College, 31-008 Krakow, Poland

**Keywords:** single ventricle, Fontan operation, cardiac arrhythmias, catheter ablation

## Abstract

Structural, hemodynamic, and morphological cardiac changes following Fontan operation (FO) can contribute to the development of arrhythmias and conduction disorders. Sinus node dysfunction, junction rhythms, tachyarrhythmias, and ventricular arrhythmias (VAs) are some of the commonly reported arrhythmias. Only a few studies have analyzed this condition in adults after FO. This study aimed to determine the type and prevalence of arrhythmias and conduction disorders among patients who underwent FO and were under the medical surveillance of the John Paul II Hospital in Krakow. Data for the study were obtained from 50 FO patients (mean age 24 ± 5.7 years; 28 men (56%)). The median follow-up time was 4 (2–9) years. Each patient received a physical examination, an echocardiographic assessment, and a 24 h electrocardiogram assessment. Bradyarrhythmia was diagnosed in 22 patients (44%), supraventricular tachyarrhythmias in 14 patients (28%), and VAs in 6 patients (12%). Six patients required pacemaker implantation, and three required radiofrequency catheter ablation (6%). Arrythmias is a widespread clinical problem in adults after FO. It can lead to serious haemodynamic impairment, and therefore requires early diagnosis and effective treatment with the use of modern approaches, including electrotherapy methods.

## 1. Introduction

Patients with congenital heart disease with single ventricle physiology constitute a heterogeneous and difficult-to-treat group, both in terms of hemodynamic changes and rhythm and conduction disorders. Currently, the preferred therapy for this patient population is the Fontan operation (FO), which aims at separating pulmonary and systemic circulation in order to achieve normal or near-normal blood oxygenation [1]. Over the last 50 years, the methods and techniques used for the surgical treatment of univentricular heart defects have undergone numerous changes [2]. Initially, FO was performed in a single stage, connecting the right atrium with the pulmonary artery (known as atriopulmonary connection (APC)) either directly or using a vascular prosthesis. However, this approach was eventually abandoned due to the high mortality rate and high number of complications. APC has now been replaced with the total cavopulmonary connection (TCPC), which includes an intracardiac or extracardiac conduit (ECC) between inferior vena cava and pulmonary artery, with bidirectional Glenn connection (superior vena cava–pulmonary artery) [1,2]. Some patients undergo fenestration, which allows unsaturated blood to be shunted to the systemic circulation at the atrium level, increasing cardiac output at the cost of cyanosis [3]. However, surgical traumatization, remodeling due to enhanced atrioventricular regurgitation, and increased stress and fibrogenesis of atrium and ventricle walls leads to the formation of areas of nonhomogeneous electric activity which in turn contributes to the development of arrhythmia and conduction defects [4].

Arrhythmias commonly observed in patients who, following FO, include sinus node and atrio-ventricular dysfunction, atrial tachycardia (ranging from focal arrhythmia to macro-reentry involving conduit as well as systemic atrium), atrial flutter (AFL), atrial fibrillation (AF) and ventricular arrhythmias (VAs) [1]. Thus far, only a few studies have analyzed the incidence of arrhythmic events in patients following FO. According to the literature, the incidence of sinus node dysfunction ranges from 9 to 60% [5,6,7,8,9,10]. In a study on a group of pediatric patients, Stephenson et al. [11] observed that 9.4% of children had supraventricular tachycardia (SVT) and 3.5% had VA after 8.6 years of FO. A Danish study conducted among adults >20 years of age reported that the incidence of arrhythmias in population following FO was 32% [12].

Treatment of the arrythmias in this group of patients due to complex anatomy, multiple surgical scars, and hypertrophied atrium could be very challenging. Pharmacotherapy is often ineffective and may lead to significant bradycardia. Catheter ablation can be safe and effective in reducing arrythmia burden after FO but require experienced electro-physiologist and well-equipped medical facilities [13].

The occurrence of arrhythmias significantly increases the risk of developing heart failure and sudden cardiac death (SCD) [1]. Therefore, early identification of patients who are at the highest risk of death due to arrhythmias (or atrial arrhythmias with rapid conduction to ventricles) following FO is important. 

In this study, we aimed to determine the type and prevalence of rhythm and conduction abnormalities and present our own experience in treating patients following FO who were under the medical supervision of the Congenital Heart Disease Team of the John Paul II Hospital in Krakow.

## 2. Materials and Methods

### 2.1. Study Participants

This retrospective study included 50 adult patients aged over 18 years. All patients underwent FO following the diagnosis of a functionally single ventricular heart and were under the care of the John Paul II Hospital. The main exclusion criteria included diabetes, current infection, inflammation, neoplastic disease, major trauma, pregnancy, and history of alcohol abuse. The median (Q1–Q3) follow-up time was 4 (2–9) years. During regular visits in our institution, each patient went through all required examinations and procedures. Depending on their results, specific therapeutic decisions were made. The demographic, anatomic, and clinical data of the patients were obtained from their medical records. Each patient was subjected to a physical examination and an assessment of the body mass index (BMI), systemic ventricle ejection fraction, and arterial oxygen saturation. BMI was calculated by dividing the weight of the patient (kg) by height (m^2^). 

### 2.2. Echocardiography

Ejection fraction of the single ventricle was assessed using Simpson’s method. Valvular competence was also evaluated in the patients by two experienced, independent cardiologists using echocardiography (Vivid 7, GE Medical Systems, Milwaukee, WI, USA) as previously described [14]. 

### 2.3. Ambulatory 24-h Electrocardiogram

Standard 24-h electrocardiographic monitoring was performed using commercially available Holter systems in all patients during their daily activities. All Holters were reviewed by two experienced observers. Recordings were analyzed using a PC-based Holter system, and those shorter than 21 h were excluded from the assessment. The predominant rhythm was defined as the one present for >50% of the time during the Holter recording.

### 2.4. Arrhythmia

The following groups of rhythm abnormalities were defined as tachyarrhythmias: (1) SVT, including sustained and non-sustained atrial tachycardia (AT, nsAT), supraventricular ectopic beats (Svebs), and AFL and AF; (2) VAs including sustained ventricular tachycardia (VT) and non-sustained ventricular tachycardia (nsVT), premature ventricular contraction (PVC), and ventricular fibrillation (VF). SVT and VAs were defined based on the guidelines of the European Society of Cardiology (ESC) [15,16,17]. Bradyarrhythmia was defined as heart rate < 60 beats per minute for a minimum of 1 min and included sinus node dysfunction and atrioventricular block (AVB) [18]. 

### 2.5. Mapping and Ablation Procedure

Patients with sustained, symptomatic atrial tachyarrhythmias, who had failed medical treatment, were recommended to undergo ablation. Catheter ablation procedures were performed under general anesthesia using the CARTO electroanatomic mapping system (Biosense Webster Inc., Diamond Bar, CA, USA). In the first case with SVT and VA ablation (2010, 2013), mapping was performed using an ablation catheter, and in the second case (2021) using a multipolar Pentaray diagnostic catheter supported with the Coherent Mapping algorithm (CARTOPRIME module). In two cases (2010, 2013), ablations were performed using 3.5 mm catheters with an open irrigated tip (Biosense Webster Navistar Thermocool), and in one case (2021) using 3.5 mm catheters with an open irrigated tip and a contact force sensor (SF, Thermocool SmartTouch^®^, Biosense Webster Inc., Diamond Bar, CA, USA). The parameters used for radiofrequency (RF) application were as follows: power: 35 W, flow of irrigation: 15–30 mL/min, time of application: 60 s, temperature limit: maximum 45 °C, and ablation index: posterior wall—400 and the other side—500. If arrhythmias were not present at the onset of the procedure, voltage maps (0.1–0.3 mV) were collected during sinus rhythm (Figure 1a), followed by which arrhythmias were induced with programmed atrial pacing from s 10-pole steerable diagnostic catheter located in the lateral tunnel (LT). The arrythmia mechanism was determined by high-density activation mapping and entrainment pacing (if possible) (Figure 1b). The origin of PVC was identified by activation mapping, in which the earliest endocardial potential advancing QRS was determined during PVC. Additionally, pace-mapping was applied in the earliest activation spot to confirm the localization of PVC, with a PVC compatibility of at least 95% of the complexes analyzed by an electrophysiological recording system (LABSYSTEM™ PRO, Boston Scientific, Boston, MA, USA). The SVT ablation protocol assumed initial mapping of LT followed by pulmonary atrium access if arrythmia elimination required this. In one case of nsVT or PVC, a retrograde approach was used. The minimum follow-up period was 12 months.

Long-term efficacy was defined as the absence of SVT symptoms or arrhythmia episodes recorded in electrocardiogram (ECG) (or in 24 h electrocardiographic monitoring). In cases with VA, it was defined as a significant reduction in arrhythmia (>80% reduction in the initial amount) or the lack of VT or nsVT observation in ECG or in 24 h electrocardiographic monitoring after a healing period (after 3 months) or in repeated 24 h electrocardiographic monitoring (every 6–12 months).

### 2.6. Statistical Analysis

Data were presented as numbers and percentages for categorical variables, means with standard deviations (SDs) for normally distributed continuous variables, and medians with lower and upper quartiles (Q1–Q3) for continuous variables with a nonnormal distribution. The normality of data distribution was verified by a Kolmogorov–Smirnov test. Categorical variables were analyzed using the χ^2^ test or Fisher’s exact test as appropriate. All the analyses were performed in IBM SPSS Statistics for Windows, Version 25.0 (IBM Corp., Armonk, NY, USA). Statistical significance was set at *p* < 0.05.

## 3. Results

### 3.1. Patients’ Characteristics

A total of 50 adult patients (mean (SD) age: 24 (5.7) years; 28 men (56%)) who underwent FO were enrolled in the study. The median (Q1–Q3) age of patients at the time of surgery was 4 (2–6) years, and the mean (SD) time after surgery was 20.5 (4.7) years. Out of 50 patients, 34 (68%) had fenestration, while 16 (32%) had no fenestration. The mean ejection fraction of the systemic ventricle was 53 ± 9.9%. The baseline characteristics of the studied patients are presented in Table 1.

### 3.2. Arrhythmia

Among the studied patients, bradyarrhythmia was detected in 25 (50%). The most common arrhythmia observed in patients after FO was sick sinus syndrome, which was primarily symptomatic and caused nocturnal bradycardia up to 35/min. Holter records of six patients showed pauses over 2 s. Low-atrial rhythm was recorded in five patients, and atrioventricular dissociation with substitute nodal rhythm in five. The AVB type 1 was observed in two patients, while advanced AVB (type 2 or 3) occurred in six patients and required pacemaker implantation for permanent pacing. Five devices were implanted by cardiac surgery (epicardial single (ventricle) chamber pacemaker electrode), and one was implanted intravenously (dual-chamber pacemaker) (Figure 2). After implantation, two patients developed cardiac device-related infective endocarditis requiring the removal and reimplantation of the device.

Supraventricular tachyarrhythmias were noted in 14 patients (28%), of which three patients had AT (6%) and one additionally had paroxysmal AF. Permanent AF was observed in one patient. Two patients with AT (4%) required RF catheter ablation due to significant symptoms, and one had asymptomatic permanent AT with a daily average ventricular rate of about 50–55/min and remains under clinical observation. The most commonly observed arrhythmias in the SVT group were nsAT and Svebs (*n* = 8, 16%). From this group, three patients required medical treatment with B-blockers due to the presence of arrhythmia symptoms. 

VAs were observed in six patients (12%). nsVT and PVC were recorded in all these patients. None of the patients had VT or VF. One patient required RF ablation, due to symptomatic Vas. Among the six patients, two were treated with sympathico-mimetic (salbutamol) for concomitant sinus bradycardia and one required pharmacological treatment (sotalol) due to the presence of severe VAs (nsVT) and symptoms.

Prevalence of arrhythmia identified based on Holter measurements, the number of pacemakers implanted, and ablation procedures performed in the patients after FO are presented in Table 2.

### 3.3. Catheter Ablation

Ablation was performed in three patients. Two patients had paroxysmal AT, and one had nsVT and PVC. All patients were symptomatic and resistant to pharmacological treatment. In the first case with SVT, macro-reentrant AT (cycle length: 250 ms) was eliminated from pulmonary atrium (retrograde access). In the long-term follow-up, recurrence of AT and AF was observed after 2 years, and the patient required the implantation of pacemaker (concomitant bradycardia) and intensification pharmacological treatment. In the second SVT case, two ATs were induced during ablation. The first AT was eliminated in LT, and the second was located out of LT. After failure of trans-fenestration access and additionally due to clinical character of the AT (self-terminated, not repeatable induction) trans-baffle puncture was not performed. If symptomatic arrythmia episodes recur during follow up period, redoing the procedure should be considered. During follow-up, recurrence of arrhythmia was not observed in the patients. VAs, in the form of ectopic nsVT and PVC, were located in the anterior basal side of single ventricular heart. Activation mapping and pace-mapping were carried out to determine the origin of arrhythmia. Acute success was achieved after RF application. Recurrence of arrhythmia was not observed in long-term follow-up. This patient died 6 years after ablation. 

### 3.4. Influence of Systemic Ventricle Morphology and Fenestration on the Incidence of Rhythm Abnormalities

The presence of the right ventricle in the place of a systemic ventricle was associated with a higher risk of developing ventricular rhythm abnormalities (Table 3, Figure 3a). 

No relationship was found between the presence of fenestration and the incidence rate of rhythm and conduction abnormalities (Table 3, Figure 3b).

This study did not compare the influence of the type of surgery conducted (APC vs. TCPC) on the incidence rate and type of rhythm abnormalities as the studied population included a low number of patients who underwent APC (only 2 patients, i.e., 4%).

### 3.5. The Survival Assessment of Enrolled Patients

One patient, a 33-year-old man, died during the study, which is 2% of the study group and 2.9% of the patients with arrhythmias. He had undergone catheter ablation due to VAs. Recurrence of arrhythmia was not observed in the 6-year follow-up period. The reason for death was a serious hemodynamic impairment in the course of significant atrioventricular regurgitation and other extra-cardiac complications including hepatic disorders.

## 4. Discussion

In this study, we determined the type and prevalence of arrhythmias in the group of adult people after FO. 

Among the studied patients, bradyarrhythmia was detected in 25 (50%), of which six (12%) required pacemaker implantation. Sinus node dysfunction may be caused by primary anatomical changes (occurring) in the single ventricular heart or due to direct damage to its vascular supply [19]. Previous works that compared the incidence of sinus node dysfunction in patients following LT conduit and ECC do not clearly indicate which surgical method can lower the risk of conduction abnormalities [5,7,8,20,21]. However, the latest guidelines regarding congenital heart diseases in adult patients support the use of ECC [1]. Symptomatic sinus node dysfunction requires permanent pacemaker implantation. In patients after FO, electrodes are located on the epicardium. Thus, every single decision regarding implantation, particularly in the case of young patients, should be made with caution, taking into account the possible complications in follow-up observation [22]. 

In the present study, supraventricular arrythmias were observed in 14 patients (28%). These data are comparable with an available study which reported that supraventricular tachyarrhythmias (including typical intra-atrial reentrant tachycardia, AFL, AF, and focal AT) were observed in approximately 20% of patients after 10 years of FO [23]. The incidence rate of arrhythmias was lower after TCPC than after APC and also lower after ECC than after intracardiac connection (LT) [1]. Studies in the literature have described several independent predictors of SVT, including the condition following APC, preoperative SVT, elderly age at the time of FO and observation period, thromboembolic episode, pacemaker implantation, moderate/serious atrioventricular valve regurgitation, and atrial enlargement [12,23]. Supraventricular arrhythmias, along with rapid conduction to ventricles, may aggravate hemodynamics in Fontan circulation and lead to heart failure over a short period of time and, in the worst cases, SCD [24]. Patients with supraventricular arrhythmias should be referred for electrophysiological assessment as soon as possible, and ablation should be performed if possible [1,25]. According to the literature, the efficacy rate of ablation is 50–70% [26]. In the present study, ablation was performed in two patients (4%). In the long-term follow-up, in the first patient recurrence of AT and AF was observed after 2 years and the patient required the implantation of a pacemaker (concomitant bradycardia) and intensification pharmacological treatment. 

In the second patient, no recurrence of SVT was observed in the 8-month observation period.

Ventricular rhythm abnormalities were recorded in six of the studied patients (12%). In our studied patients the only form of VAs was ns VT and PVC and none of the patients developed sustained VT or VF. In a study by Stephenson et al. [11], the incidence of VA after 8.6 years of FO was 3.5% of patients. In a study on the combined pediatric and adult population by Rychik et al. [27], the incidence rate of ventricular rhythm abnormalities, including SCD, was assessed at 2–10%. The mechanism contributing to the formation of VAs in the population of patients following FO remains unknown. It is assumed that VAs may form from a ventriculotomy scar which allows for reentry circulation, enlargement of the defect in the interventricular septum, increase in myocardial fibrosis, and ventricular dilatation. These are especially dangerous for patients with anatomical right systemic ventricle and single-ventricle dysfunction [11]. In this study, we observed that patients with anatomical right systemic ventricle were more likely to have an increased incidence of ventricular rhythm abnormalities. 

Episodes of nsVTs were recorded in six patients. One of these patients, a 27-year-old man, who was ineffectively treated with antiarrhythmic drugs, was qualified for and received RF ablation, the effect of which was sustained in the 6-year follow-up period. In recent years, novel treatment methods for arrhythmias have been developed, including ablation, and electrophysiologists and cardiologists cooperate better in treating patients with congenital heart defects. Furthermore, the development of electrophysiological mapping systems has increased the possibilities of mapping and enabled the ablation of complex atrial arrhythmias [28].

Malignant VAs may lead to SCD. Khairy et al. [29] stated that the incidence of SCD in distant time from FO is 0.15% and in most of the cases reported annually, SCD is caused by arrhythmias, although it is difficult to identify the possible prognostic factors. The treatment of VAs using an implantable cardioverter defibrillator (ICD) in patients after FO is a major challenge for electrocardiologists. The available studies in the literature have described only single patients who required treatment with ICD after FO [30,31,32,33]. In the present study, there were no cases in the patient group that would require ICD implantation. 

With the extension of follow-up time, the number of arrhythmias and conduction disorders may increase significantly. Thus, the differences in the frequency of their occurrence described in the literature may result from the length of the observation time. 

## 5. Conclusions

Early diagnosis of arrhythmia in the patients who underwent FO is important as arrhythmias may contribute to heart failure, embolization, and SCD. These patients should be treated in multidisciplinary centers by experienced specialists with the use of modern methods, such as ablation, which is a safe and effective method for treating both supraventricular arrhythmias and VAs. The treatment of this patient group with ICD still remains a huge challenge for electrocardiologists. 

Further analyses on larger groups of patients are necessary to create multi-center registries. The experience exchange between centers is required for the development of guidelines for treating this complex group of patients. 

### Limitation of the Study

The limitation of this retrospective study is that the number of patients in the study group was small and the population was relatively mixed. As the Fontan procedure was introduced into clinical practice relatively recently, we can expect that single centers at this stage of medical development may have fewer patients under medical supervision. Thus, the limitation in the number of patients in the study group applies to most of the individual centers in the world caring for this patient population.

## Figures and Tables

**Figure 1 jcm-11-01968-f001:**
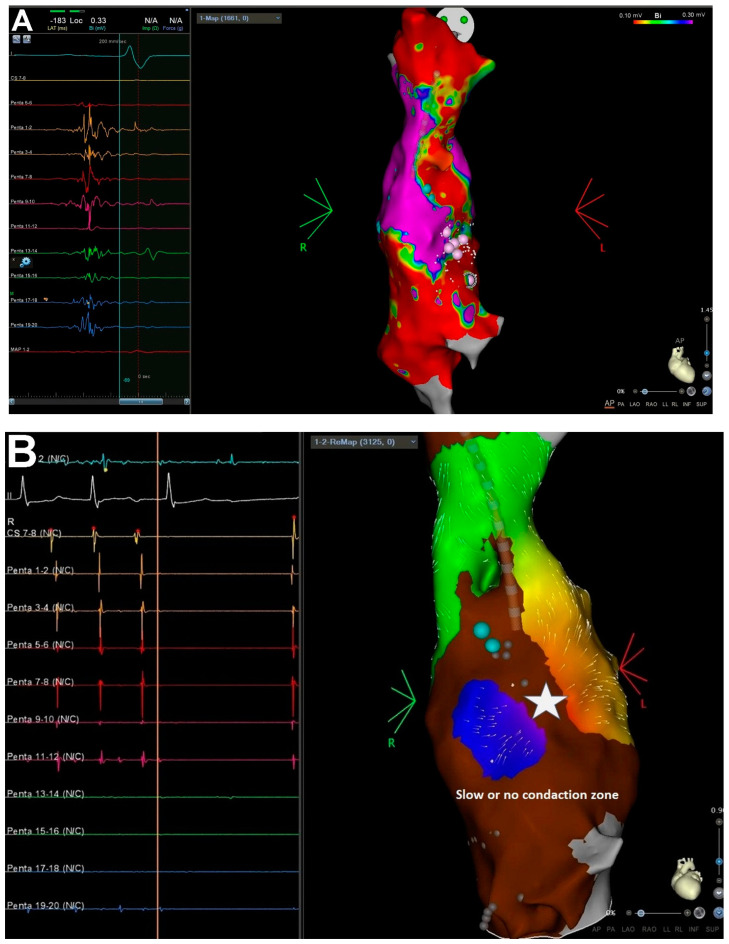
Procedure of supraventricular arrhythmia ablation in a patient after FO. (**A**) Fast anatomic map with a high-density voltage map (0.1–0.3 mV) of LT collected during sinus rhythm (CARTO, Biosense Webster Inc.). Extensive scaring at the anterolateral area of the tunnel with multiple double (blue dots) and fragmented potentials (white dots). Fragmented potentials recorded from Pentary diagnostic catheter are shown on the left side of the figure. (**B**) Activation map of AT (cycle length (CL): 380 ms, 80% of the CL in LT) created with a coherent mapping algorithm (CARTOPRIME, Biosense Webster Inc.). Electrocardiograms recorded from ablation and diagnostic catheters during AT termination are shown on the left side of the figure. White star—spot of AT termination during radiofrequency application.

**Figure 2 jcm-11-01968-f002:**
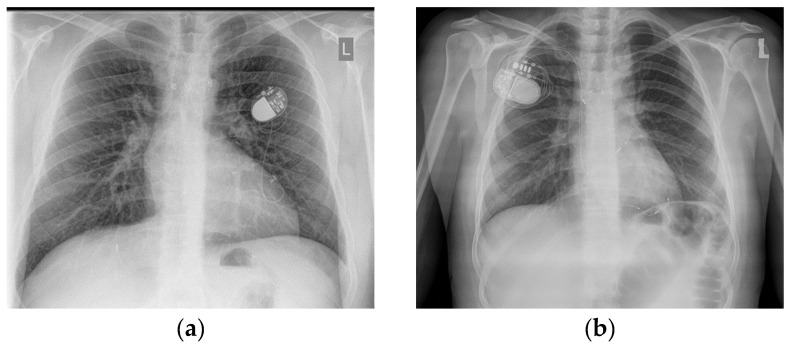
Chest X-ray after pacemaker implantation in atriopulmonary projection after an FO procedure: (**a**) with a VVI epicardial lead after a cardiosurgery procedure and (**b**) with a DDD endocardial lead after transvenous implantation. L means left side.

**Figure 3 jcm-11-01968-f003:**
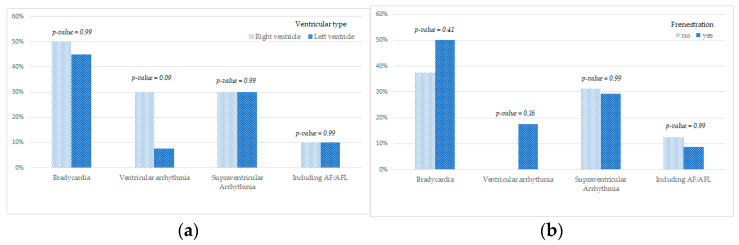
Dependency between ventricular type (**a**) and fenestration (**b**) and the incidence of rhythm abnormalities.

**Table 1 jcm-11-01968-t001:** Baseline characteristics of patients after Fontan operation (FO).

Variables	Fontan Patients (*n* = 50)
Age, years	24 (5.7)
Female sex, *n* (%)	22 (44)
Height, cm	170 (8.1)
Body mass index, kg/m^2^	22.6 (3.2)
Anatomic diagnosis, *n* (%)	
Tricuspid atresia	8 (16)
Pulmonary stenosis/TGA	15 (30)
Right ventricular hypoplasia	13 (26)
Hypoplastic left heart syndrome	6 (12)
Double-outlet right ventricle with left ventricular hypoplasia	6 (12)
Double-inflow left ventricle	1 (2)
Common atrioventricular canal	1 (2)
Systemic ventricle type, *n* (%)	
Left ventricle	30 (60)
Right ventricle	20 (40)
NYHA functional class, *n* (%)	
I	5 (10)
II	41 (82)
III	4 (8)
IV	0 (0)
Types of FO, *n* (%)	
Total cavopulmonary connection, 48 (96)	
Lateral tunnel	47
Extracardiac conduit	1
Atriopulmonary connection	2 (4)

Abbreviations: NYHA, New York Heart Association; TGA, transposition of great arteries. Continuous data are presented as means (SD), and categorical data as numbers (percentage).

**Table 2 jcm-11-01968-t002:** Holter measurements, the number of pacemakers implanted, and ablation procedures performed in patients after FO.

Arrythmia Type, Catheter Ablation, Device Implanted	Fontan Patients (*n* = 50)
Dominant SSS with bradycardia	25 (55%)
- Pause >2 s	6
- Low-atrial rhythm	5
- Nodal rhythm/atrioventricular dissociation	5
- AVB-1	2
- AVB-2	0
- AVB-3	6
Supraventricular tachyarrhythmias	14 (28%)
- In the form of sustained AT	3 (6%)
- In the form of nsAT, Svebs	8 (16%)
- In the form of AF/AFL	2 (4%)
VAs (in the form of nsVT and PVC)	6 (12%)
Catheter ablation	3 (6%)
- Paroxysmal AT	2 (4%)
- nsVT and PVC	1 (2%)
Device implanted (VVI/DDD)	6 (12%)
VVI—5, DDD—1, 2 devices removed because of cardiac device-related infective endocarditis	

Abbreviations: SSS, sick sinus syndrome; AVB-1, atrioventricular block type 1, AVB-2 atrioventricular block type 2; AVB-3, atrioventricular block type 3; AT, atrial tachycardia; nsAT, non-sustained atrial tachycardia; Svebs, supraventricular ectopic beats; AF, atrial fibrillation; AFL, atrial flutter; VAs, ventricular arrhythmias; nsVT, nonsustained ventricular tachycardia; PVC, premature ventricular contraction; VVI, single (ventricle) chamber pacemaker; DDD, dual-chamber pacemaker.

**Table 3 jcm-11-01968-t003:** Dependency between fenestration and systemic ventricle morphology and incidence of rhythm abnormalities.

	Bradycardia				Ventricular Arrhythmia				Supraventricular Arrhythmia				Including AF/AFL			
	no		yes		no		yes		no		yes		no		yes	
Fenestration	*n*	%	*n*	%	*n*	%	*n*	%	*n*	%	*n*	%	*n*	%	*n*	%
no	10	62.50%	6	37.50%	16	100%	0	0.0%	11	68.8%	5	31.3%	14	87.5%	2	12.5%
yes	17	50.00%	17	50.00%	18	82.4%	6	17.6%	24	70.6%	10	29.4%	31	91.2%	3	8.8%
*p*-value	0.41				0.16				0.99				0.99			
	no		yes		no		yes		no		yes		no		yes	
Ventricular type	*n*	%	*n*	%	*n*	%	*n*	%	*n*	%	*n*	%	*n*	%	*n*	%
Right ventricle	5	50.0%	5	50.0%	7	70.0%	3	30.0%	7	70.0%	3	30.0%	9	90.0%	1	10.0%
Left ventricle	22	55.0%	18	45.0%	37	92.5%	3	7.5%	28	70.0%	12	30.0%	36	90.0%	4	10.0%
*p*-value	0.99				0.09				0.990				0.99			

## Data Availability

The data presented in this study are not publicly available due to upcoming publications but are available on request from the corresponding author.

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
