# Peer review of "Prevalence of Arrhythmia in Adults after Fontan Operation"

_jcm, 2022, doi:10.3390/jcm11071968_

Round 1
Reviewer 1 Report
Title: “The prevalence of arrhythmia in adults after Fontan operation”
Dear Authors,
I read this manuscript and I think that:
- It would have been interesting to consider a control group of healthy patients and to compare the incidence /prevalence of different types of arrhythmias in these two groups. Please discuss such a point.
- The small sample size – although the prevalence of the disease is limited – might be considered as a limitation of the study. This should be discussed in a dedicated limitation section.
Reviewer 2 Report
The authors of the current manuscript present results from an original study, assessing the prevalence and type of rhythm and conductance disorders of patients who underwent Fontan operation.
I have the following recommendation to the authors (most of them - technical):
In the title - just “Prevalence” without “The”
In the Abstract: “Structural, hemodynamic, and morphological changes occurring in the heart of an adult patient following Fontan operation (FO) can contribute… - I suggest “Structural, hemodynamic, and morphological cardiac changes following Fontan operation (FO) can contribute…”
Line 24-25: “…medical supervision…” should be “medical surveillance”
Line 30-34: The text there must be modified to a more appropriate conclusion observing the title and the results, presented in the abstract.
Line 39-40: “hemodynamic disorders and rhythm and conduction defects.” could be “hemodynamic changes (or impairment) and rhythm and conduction disorders.”
Please, add citations to the first half of the text in the Introduction
Reviewer 3 Report
The paper present a case study report about the prevalence of arrhythmia in adults after the Fontan operation. The content of the paper is sufficient for journal publication, however, the presentation of the paper needs an improvement.
- In my opinion, an Introduction section can be added with more content.
- Figure 2. The ECG signal in Figs. 2(a) and (b) were not obvious. I suggest enlarging the Figures.
- Please tidy up the presentation of Table 3 as the Table title is missing and located on a different page.
- It also would be good, if a Figure can be presented according to Table 3 for better visualization of the data.
- Since the Authors, provided a Limitation in the Conclusion section, I suggest the Authors also provide future work information in the Conclusions section.
